# Light-induced unfolding and refolding of supramolecular polymer nanofibres

Bimalendu Adhikari[1,2], Yuki Yamada[1], Mitsuaki Yamauchi[1], Kengo Wakita[1], Xu Lin[1], Keisuke Aratsu[1], Tomonori Ohba[3], Takashi Karatsu[1], Martin J. Hollamby[4], Nobutaka Shimizu[5], Hideaki Takagi[5], Rie Haruki[5], Shin-ichi Adachi[5] & Shiki Yagai[1]

Unlike classical covalent polymers, one-dimensionally (1D) elongated supramolecular polymers (SPs) can be encoded with high degrees of internal order by the cooperative aggregation of molecular subunits, which endows these SPs with extraordinary properties and functions. However, this internal order has not yet been exploited to generate and dynamically control well-defined higher-order (secondary) conformations of the SP backbone, which may induce functionality that is comparable to protein folding/unfolding. Herein, we report light-induced conformational changes of SPs based on the 1D exotic stacking of hydrogen-bonded azobenzene hexamers. The stacking causes a unique internal order that leads to spontaneous curvature, which allows accessing conformations that range from randomly folded to helically folded coils. The reversible photoisomerization of the azobenzene moiety destroys or recovers the curvature of the main chain, which demonstrates external control over the SP conformation that may ultimately lead to biological functions.

[1] Graduate School of Engineering, Chiba University, 1-33 Yayoi-cho, Inage-ku, Chiba 263-8522, Japan. [2] Department of Chemical Sciences, Indian Institute of Science Education and Research (IISER) Mohali, Knowledge City, Sector 81, S.A.S. Nagar, Manauli PO, Punjab 140306, India. [3] Graduate School of Science, Chiba University, 1-33 Yayoi-cho, Inage-ku, Chiba 263-8522, Japan. [4] School of Chemical and Physical Sciences, Keele University, Keele, Staffordshire ST55BG, UK. [5] Photon Factory, Institute of Materials Structure Science, High Energy Accelerator Research Organization, Tsukuba 305-0801, Japan. Correspondence and requests for materials should be addressed to S.Y. (email: yagai@faculty.chiba-u.jp).

One-dimensionally (1D) elongated molecular assemblies with high degrees of internal order[1], categorized as supramolecular polymers (SPs) (ref. 2), represent emerging stimuli-responsive materials with interesting functions and properties[3–8]. A variety of functional SPs have been developed, while precise supramolecular polymerization has been devised recently, which is inspired by the biological system as well as synthetic polymers[9,10]. An important issue that has not yet been addressed in the context of SPs is dynamic control over their higher-order structures at the nanoscale (for example, helical coils), despite the fact that typical helical columnar structures at the primary level[11–15] have been well studied in terms of internal order. Changes in the higher-order conformation of biopolymers play an important role in biological systems, which is arguably best reflected in protein folding[16]. For instance, folded proteins are functional, whereas unfolded/ misfolded proteins generally lose their functionality. Inspired by these naturally occurring polymers, the control over folding has been explored in synthetic covalent polymers[17–22] and oligomers[21,23–26], which are referred to as foldamers[21,27–29]. In order to mimic the functions of biopolymers and/or compete with oligomeric as well as polymeric foldamers, remote control over the conformation of the SP backbone is required.

To address this issue, herein, we have used a previously reported barbiturated naphthalene derivative, which polymerizes non-covalently into uniform toroidal nanostructures via stacking of hydrogen-bonded hexamers[30,31]. The introduction of a photoresponsive azobenzene into the parent-barbiturated naphthalene results in increased attractive forces between the stacking molecules, and causes light-induced changes of the molecular shape[32–36]. Depending on the polymerization conditions, the new molecule non-covalently polymerizes into various quasi 1D structures with spontaneous curvature, ranging from randomly coiled to helically folded fibres. Photoisomerization of the azobenzene unit by exposure to ultraviolet (UV) or visible (Vis) light provides a means to reversibly destroy or recover the curvature of the main chain, respectively. Thus, an external dynamic control over helical secondary structures of SPs is demonstrated. Although the terminologies conformation and folding/unfolding are commonly reserved for covalent polymer chains, we are using them here in the context of non-covalent polymer chains to illustrate the conceptual similarity.

## Results

**Molecular design.** For the construction of well-defined helically coiled conformations, the design of foldamers is usually based on covalent local constraints, which manifest in covalently curved units that typically arise from *meta*-isomers of aromatics (Fig. 1a)[21,23–26]. As these helically coiled structures are generally stabilized by non-covalent interactions between non-adjacent monomer units (interloop interactions), external stimuli afford dynamic control over helically coiled and extended/randomly coiled conformations[21,23–25]. As an alternative strategy, Hecht and co-workers introduced azobenzene moieties into the backbone of foldamers to realize direct folding and unfolding by light[21,22,26]. Exploiting a similar strategy in order to gain dynamic conformational control over SPs with a high degree of internal order thus initially requires non-covalently curved stacks with shape persistence (Fig. 1b). Moreover, the main chain of the SPs should be amendable to dynamic tuning of the internal order, that is, the ability to switch on/off the non-covalent curvature (Fig. 1b), as—unlike typical foldamers—the main chains of SPs are generally not strong enough to allow for a selective dissociation of the non-covalent interactions between non-adjacent monomer units by external stimuli.

To verify this hypothesis in this study, we have used our previously reported barbiturated naphthalene molecule **1** (Fig. 2a), which self-assembles into uniform toroidal nanostructures[30,31] that can serve as non-covalently curved stacks to realize target helically folded SP systems with dynamically controllable conformations. In methylcyclohexane (MCH), the parent molecule **1** assembles into hydrogen-bonded cyclic hexamers that stack quasi 1D via π–π interactions to form uniform toroidal nanostructures (SP$_{toroid}$, Fig. 2b,c). The shape persistency of SP$_{toroid}$ and their uniformity of the diameter ($14 \pm 0.1$ nm) implies that the spontaneous curvature comprising non-covalently curved stacks occurs by highly idiosyncratic internal order[7] within the continuous hexamer stacking. The cohesive forces between these hexamers should thus be strengthened by introducing additional aromatic moieties into **1**, which could prevent the growing polymer from closing into discrete shorter assemblies, instead yielding extended and robust SPs (ref. 4) with well-defined conformations due to non-covalently curved stack. As this curvature-inducing internal order is likely due to the complex pinwheel architecture of the hexamers (Fig. 2b,c)[30], any deformation should enable a modulation of the polymer conformation by changing the internal order of the non-covalently curved stacks. This strategy may allow access to a unique class of SPs with controllable conformations.

Accordingly, we designed and synthesized molecule **2** (Supplementary Methods) wherein a photoresponsive azobenzene is introduced into the parent molecule in order to increase attractive forces between the stacking molecules, as well as cause light-induced changes of the molecular shape (Fig. 2d)[32–36]. The *trans*-isomer of **2** (*trans*-**2**), containing an additional planar π-surface, should strengthen the cohesive forces between the hexamers (Fig. 2e,f). This assumption is justified by the results of temperature-dependent UV–Vis spectroscopy measurements (Fig. 2g), which reveal a higher elongation temperature ($T_e$) and a larger enthalpy change in the elongation regime ($\Delta H_e$) of the cooperative nucleation-elongation process for *trans*-**2** ($T_e = 83.6$ °C, $\Delta H_e = -108$ kJ mol$^{-1}$) relative to **1** ($T_e = 42.5$ °C and $\Delta H_e = -83.4$ kJ mol$^{-1}$) (Fig. 2h)[37,38]. The enhanced cohesive forces should reduce the photoisomerization of azobenzene moieties, resulting in the partial formation of *cis*-**2** in the stacked hexamers (Fig. 2e). This would cause the formation of defected hexamers and affect internal order, thus antagonizing the spontaneous curvature (Fig. 2f).

**Thermally obtainable SPs.** Atomic force microscopy (AFM) measurements showed that cooling a MCH solution of *trans*-**2** ($1.0 \times 10^{-4}$ M) from 90 to 20 °C without temperature control resulted in the formation of randomly folded coils (SP$_{random}$) with a defined spontaneous curvature (Fig. 3a,c; Supplementary Fig. 1a–c). To quantify these structures, the radius of curvature, $r$, evaluated by manually fitting a circle with radius $r$ along each curve, and the turning angle, $\theta$, of the polymer segment before the clockwise or counter-clockwise change of direction, were used (Fig. 3e). The similarity in the appearance of the spontaneous curvature of SP$_{random}$ ($r_{ave} = 10.4 \pm 0.2$ nm, $\theta_{ave} = 370 \pm 5°$) (Fig. 3f,g) with that of the toroids of **1** indicates that the introduction of a *trans*-azobenzene unit in **2** does not considerably affect the internal order responsible for the curvature. The formation of a very minor amount of SP$_{toroid}$ of **2** (Fig. 3c), which exhibit the same $r_{ave}$ as SP$_{random}$, suggests that the latter is formed by an extension of the polymerization beyond the toroid due to enhanced cohesive forces between hexamers.

A cooling rate of 0.1 °C min$^{-1}$ the conditions that favour a thermodynamic polymerization[39,40] resulted in the formation of spirally folded fibres (SP$_{spiral}$; $r_{ave} = 11.4 \pm 0.2$ nm, $\theta_{ave} = 510 \pm 5°$) with more organized conformations and higher degrees of

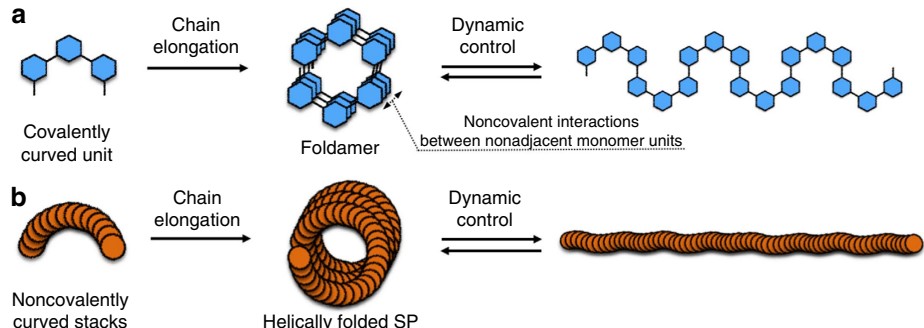

**Figure 1 | Foldamer design strategy to generate helically folded supramolecular polymers that can be controlled dynamically.** (**a**) Foldamers generally use *meta*-substituted aromatic moieties as covalently curved units. Folding/unfolding can be realized indirectly by controlling non-covalent interactions between non-adjacent monomer units, or directly by introducing responsive units (for example, azobenzene moieties) into the helical backbone/main chain. (**b**) Supramolecular polymer approach in this study: non-covalently curved stacks with high degree of internal order should be able to extend their curvature into well-defined helical conformations. Subsequent dynamic control of the thus obtained conformations should then afford direct tunability of the curvature.

internal order (Fig. 3d,f,g; Supplementary Figs 1g–l, 3, 22, Supplementary Discussion). Conversely, the spontaneous curvature was not observed upon fast quenching of the solution from 90 to 20 °C ($>100$ °C min$^{-1}$, ice bath). Instead, linear fibres ($SP_{linear}$; Fig. 3b; Supplementary Fig. 1d–f) were formed. These SPs including $SP_{random}$ share the same cross-sectional width ($\sim 9.7 \pm 0.3$ nm) and thickness ($2.7 \pm 0.2$ nm) of their backbones, suggesting that they organize via an identical self-assembly process, that is, a translationally offset stacking of hexamers (Supplementary Fig. 2)[30,31]. In order to generate a spontaneous curvature for $SP_{random}$ and $SP_{spiral}$, an additional rotational offset is required (Fig. 2f). Thus, a unique internal order, that is, the specific stacking of hexamers, that produces the spontaneous curvature should arise from a more thermodynamic supramolecular polymerization process[9,39], and the lack of spontaneous curvature in $SP_{linear}$ likely suggests a decrease of the internal order. This notion is supported by the decreased absorbance at $\lambda = 410$–$430$ nm for $SP_{linear}$ that likely measures the level of internal order (Supplementary Fig. 3). Hence, storing a quenched sample of $SP_{linear}$ for 24 h at 20 °C led to the almost complete conversion into $SP_{random}$ (Supplementary Fig. 4), which demonstrates the spontaneous recovery of a higher degree of the internal stacking order of hexamers (Fig. 3a).

This folding variation of SPs of **2** was investigated in solution by small-angle X-ray scattering (SAXS)[31]. In MCH, the SAXS plots of $SP_{random}$ and $SP_{spiral}$ exhibited increased scattering at $Q < 0.15$ nm$^{-1}$, which probably originates from the elongated polymer chains and their fractal structure (Supplementary Fig. 5a–c). Additionally, both data sets exhibit features at $Q = 0.3$–$1.2$ nm$^{-1}$, which should arise from the comparable spontaneous curvature of the SPs (*cf.* AFM results). Similar SAXS plots were observed for the toroidal nanostructures of **1** within that $Q$-range, which corresponds to a length scale of $\sim 5$–$20$ nm (ref. 31). This is hardly surprising, considering that the difference between overlapping loops (spirals/random coils) and closed toroids is relatively small at this length scale. Data fits using toroid or hollow cylinder models (Supplementary Fig. 5a–c; Supplementary Discussion) were consistent with the SAXS data and afforded curvature radii for $SP_{random}$ ($r_{ave} = 9.8 \pm 0.2$ nm) and $SP_{spiral}$ ($r_{ave} = 9.9 \pm 0.2$ nm) that agree well with the AFM values. Conversely, the SAXS data from $SP_{linear}$ retained the fractal-like scattering, but maxima/minima were not observed (Supplementary Fig. 5d). This result is indicative for the presence of an elongated polymer without spontaneous curvature (Fig. 3b), in reasonable agreement with the AFM results.

**Photochemically transmuted SPs**. Subsequently, we examined the morphological effect of the photoisomerization of the azobenzene moiety. UV–Vis and NMR spectroscopical analyses showed that irradiation of an $SP_{spiral}$ solution with UV light ($\lambda = 365$ nm) induced partial photoisomerization (23% *cis*-**2**) (Supplementary Figs 6a and 7). Despite this relatively low isomerization yield, a dynamic light scattering (DLS) analysis showed a substantial increase in average hydrodynamic diameter ($D_h$) from $\sim 320$ to $\sim 670$ nm (Fig. 4a, Supplementary Fig. 8). An increase in aggregate size upon *trans*-to-*cis* isomerization in azobenzene-based supramolecular systems is, however, counterintuitive[41], considering that the bent *cis*-isomer is generally less favourable for extended aggregation than the planar *trans*-isomer[32,42,43].

SAXS analysis of the UV-irradiated $SP_{spiral}$ solution revealed a complete disappearance of the scattering features of the spiral structures (Fig. 4b, Supplementary Fig. 9). This implies a considerable conformational change of the SP and a loss of the spontaneous curvature, caused by the *trans*-to-*cis* isomerization of the azobenzene moieties. Subsequent irradiation with Vis light ($\lambda = 470$ nm) resulted in a partial *cis*-to-*trans* isomerization ($\sim 11\%$ *cis*-**2**). Although the SAXS pattern did not recover fully (*vide infra*), the DLS analysis suggested reversible size changes upon irradiation with Vis light, which could be repeated multiple times upon alternating exposure to UV and Vis irradiation (Fig. 4a).

As expected from the SAXS and DLS results, dramatic conformational changes for $SP_{spiral}$ could also be observed by AFM and TEM after consecutive exposure to UV and Vis light. Exposure to UV irradiation transformed $SP_{spiral}$ (Fig. 4c,d) into stretched SP fibres with linear segments ($SP_{linear}$)$^{UV}$ (Fig. 4e,f), most likely by reducing the internal order as evident from the increased absorbance at $\lambda = 410$–$430$ nm (Supplementary Fig. 6b). Subsequent exposure of ($SP_{linear}$)$^{UV}$ to weak Vis light recovered most of the spontaneous curvature and internal order, affording randomly coiled SP fibres ($SP_{random}$)$^{Vis}$ (Fig. 4g–i; Supplementary Fig. 6b). An AFM analysis confirmed that ($SP_{linear}$)$^{UV}$ and ($SP_{random}$)$^{Vis}$ could be interconverted several times (Supplementary Fig. 10). Although the $r_{ave}$ values of ($SP_{random}$)$^{Vis}$ ($r_{ave} = 12 \pm 0.2$ nm; $\theta_{ave} = 267 \pm 5°$) and $SP_{random}$ ($10.4 \pm 0.2$ nm; $370 \pm 5°$) are comparable, the $\theta_{ave}$ values differ substantially (Supplementary Fig. 11a,b). This result may explain the absence of maxima/minima attributed to the presence of loops in the SAXS analysis of ($SP_{random}$)$^{Vis}$: as $\theta_{ave}$ is $<360°$, the partially open loops in ($SP_{random}$)$^{Vis}$ no longer exhibit a toroid or hollow cylinder shape, and consequently the scattering profile of ($SP_{random}$)$^{Vis}$ resembles that of the elongated state. The recovery

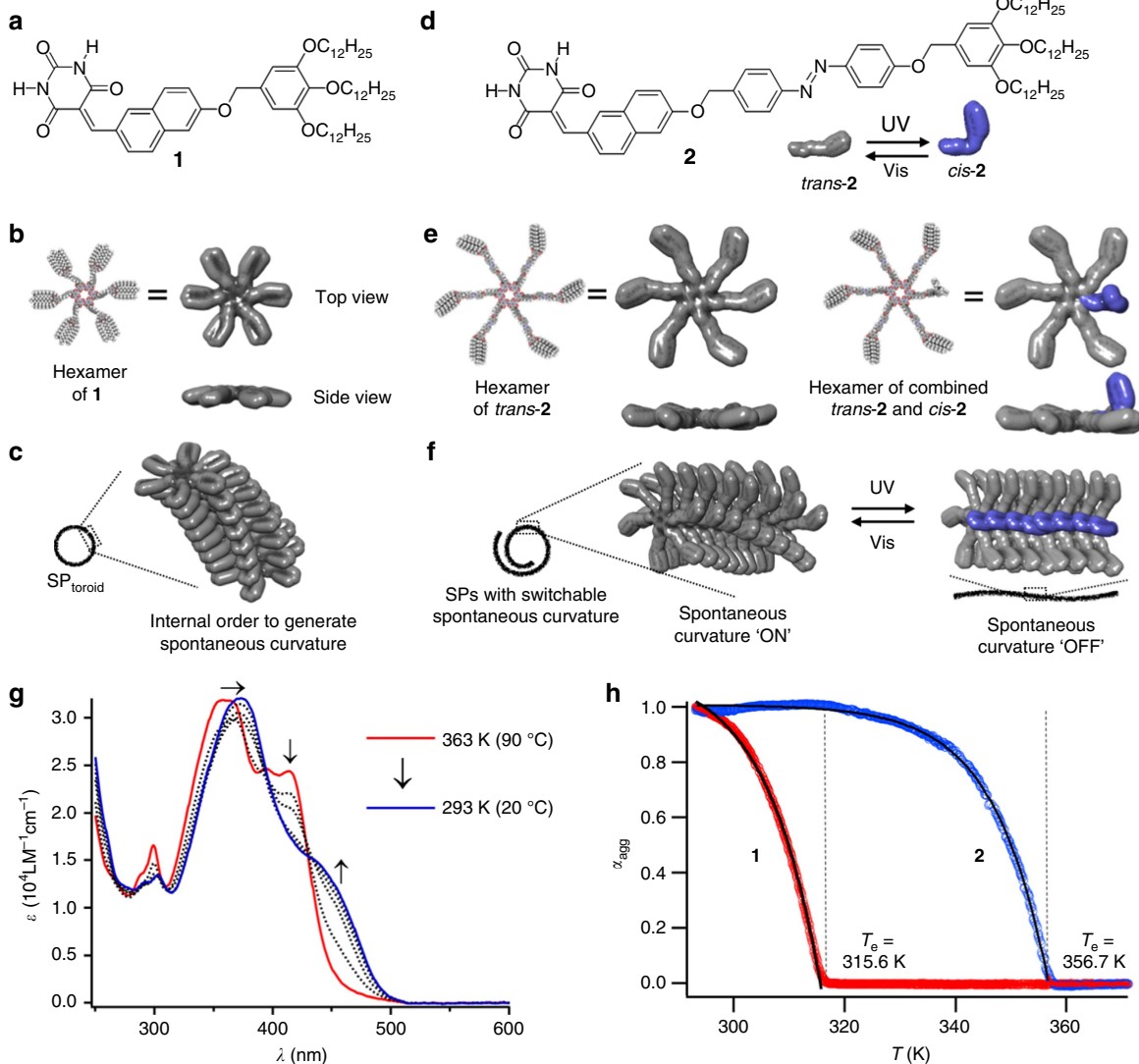

**Figure 2 | SPs with photo-tunable curvature.** (**a**) Chemical structure of previously reported **1**. (**b**) Molecular models (CPK model as well as electron cloud model) of a hydrogen-bonded hexamer of **1**. (**c**) Schematic representation for the formation of a toroidal supramolecular assembly ($SP_{toroid}$) in MCH by stacking planar hexamers of **1**. (**d**) Chemical structure of **2**, featuring a central *trans*-azobenzene moiety, which causes a large photoisomerization-induced change in its molecular shape as shown by the electron cloud model of *trans*-**2** (grey) and *cis*-**2** (blue). (**e**) Molecular models (CPK models as well as electron cloud models) of hydrogen-bonded hexamers of six *trans*-**2**, as well as of five *trans*-**2** and one *cis*-**2**. For electron cloud models, side views highlight the change of planarity of the hexamers upon partial photoisomerization of the azobenzene unit. (**f**) Schematic representation of the formation of open-ended extended SPs with photo-switchable spontaneous curvature, wherein the reversible loss and recovery of the curvature are controlled by UV and Vis light, respectively. (**g**) Temperature-dependent UV–Vis spectra of MCH solution of *trans*-**2** ($c = 2.5 \times 10^{-5}$ M, MCH) upon cooling at a cooling rate of 1 °C min$^{-1}$. Upon cooling, the $\pi$–$\pi^*$ transitions of both azobenzene (360–375 nm) and naphthalene moieties (380–430 nm to 430–500 nm) are bathochromically shifted, suggesting slipped $\pi$–$\pi$ stacking arrangement (so called *J*-type stacking) of both chromophores. (**h**) Plots of molar fractions of aggregated molecules ($\alpha_{agg}$) as a function of temperature for cooling processes of **1** ($c = 2.5 \times 10^{-5}$ M, monitored at $\lambda = 470$ nm) and *trans*-**2** ($c = 2.5 \times 10^{-5}$ M, monitored at $\lambda = 455$ nm) in MCH.

of the curvature induced by Vis light could simultaneously occur locally throughout the polymer backbone without maintaining a specific turning direction, thus resulting in randomly folded coils with a lower $\theta_{ave}$. Hence, sufficient preservation of the turning direction at each curve was achieved with a higher $\theta_{ave}$ by thermal supramolecular polymerization of the monomers.

The entanglement can be evaluated in terms of sinuosity (*S*), defined as $L/d$, wherein *d* is the shortest distance between the two ends of a given fibre with the length *L*. The value of *S* for a perfectly straight fibre (*S* = 1.0) increases upon increasing the degree of entanglement. Histograms of *L*, *d* and *S* values for randomly chosen fibres of $SP_{spiral}$, $(SP_{linear})^{UV}$ and $(SP_{random})^{Vis}$

are shown in Supplementary Fig. 12. The *L* of the above three SPs are widely distributed in the range of 100–1,500 nm, and the distribution showed no arguable difference before and after light irradiation. In contrast, the distribution of *d* displayed a great change before and after light irradiation. While the *d* of most $SP_{spiral}$ and $(SP_{random})^{Vis}$ fibres are in the range of 100–500 nm, those of $(SP_{linear})^{UV}$ fibres are distributed in a wider range up to 1,200 nm due to extension. As a result, most of the $(SP_{linear})^{UV}$ fibres have *S* values in the range of 1.0–3.0, which illustrates their extended form, whereas those of $SP_{spiral}$ and $(SP_{random})^{Vis}$ are more widely distributed in the range of 2–20, which illustrate the substantial difference in entanglement well.

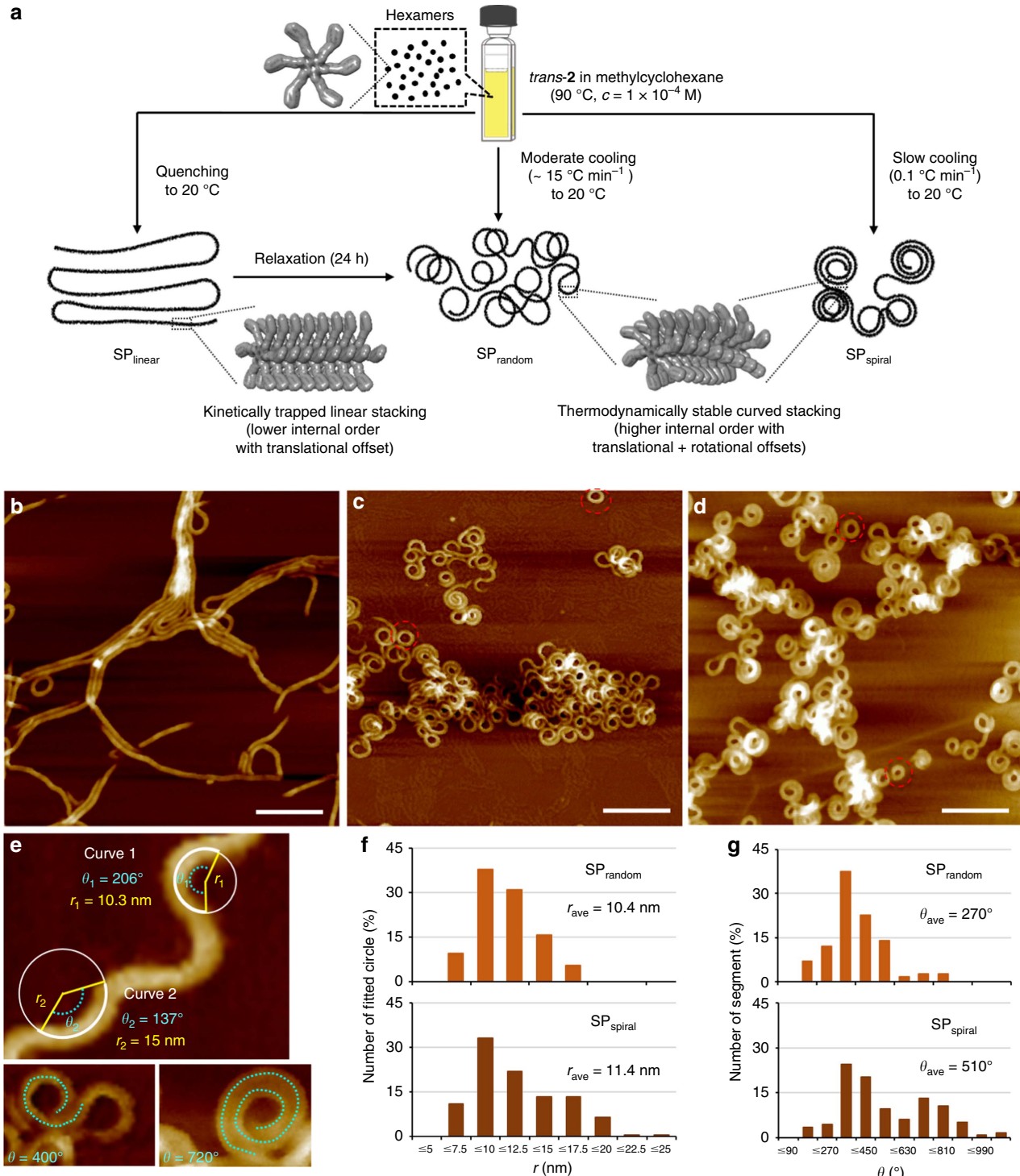

**Figure 3 | Morphological analysis of thermally obtainable SPs. (a)** Schematic representation for the preparation of three SPs ($SP_{linear}$, $SP_{random}$ and $SP_{spiral}$) with variable foldability and internal order. These SPs are prepared by cooling hot solutions of *trans-2* ($c = 1.0 \times 10^{-4}$ M) from 90 to 20 °C using three different cooling protocols. **(b–d)** AFM images (scale bar, 100 nm) of SPs of *trans-2* spin-coated from MCH solutions onto highly oriented pyrolytic graphite. **(b)** Linearly elongated SP ($SP_{linear}$) fibres were obtained from a cooling rate of $>100$ °C min$^{-1}$ (ice bath). **(c)** Randomly folded coils ($SP_{random}$) were obtained using a cooling rate of $\sim 15$ °C min$^{-1}$; these coils change direction of the curves almost at every cycle. A few nanotoroids were also observed (red dotted circles). **(d)** Spirally folded SPs ($SP_{spiral}$) fibres were obtained from a cooling rate of 0.1 °C min$^{-1}$; $SP_{spiral}$ consists predominantly of several spiral-millipede segments that are connected in a single continuous SP chain. **(e–g)** Morphological insight into $SP_{random}$ and $SP_{spiral}$. **(e)** Illustration of radius ($r$) and turning angle ($\theta$) of the spontaneous curvature in representative curved domains. **(f)** Histogram for the distribution of $r$ obtained by analyzing $>100$ curved segments. **(g)** Histogram for the distribution of $\theta$ obtained by analyzing $>100$ curved domains.

**Mechanistic Insight**. The light-induced conformational changes of SPs could occur via two different pathways, that is, direct or indirect pathways (Fig. 5a). In the direct pathway, the azobenzene moieties, that are embedded in the SP chains, isomerize and change the stacking arrangement of hexamers, thus antagonizing the spontaneous curvature. In the indirect pathway, on the other

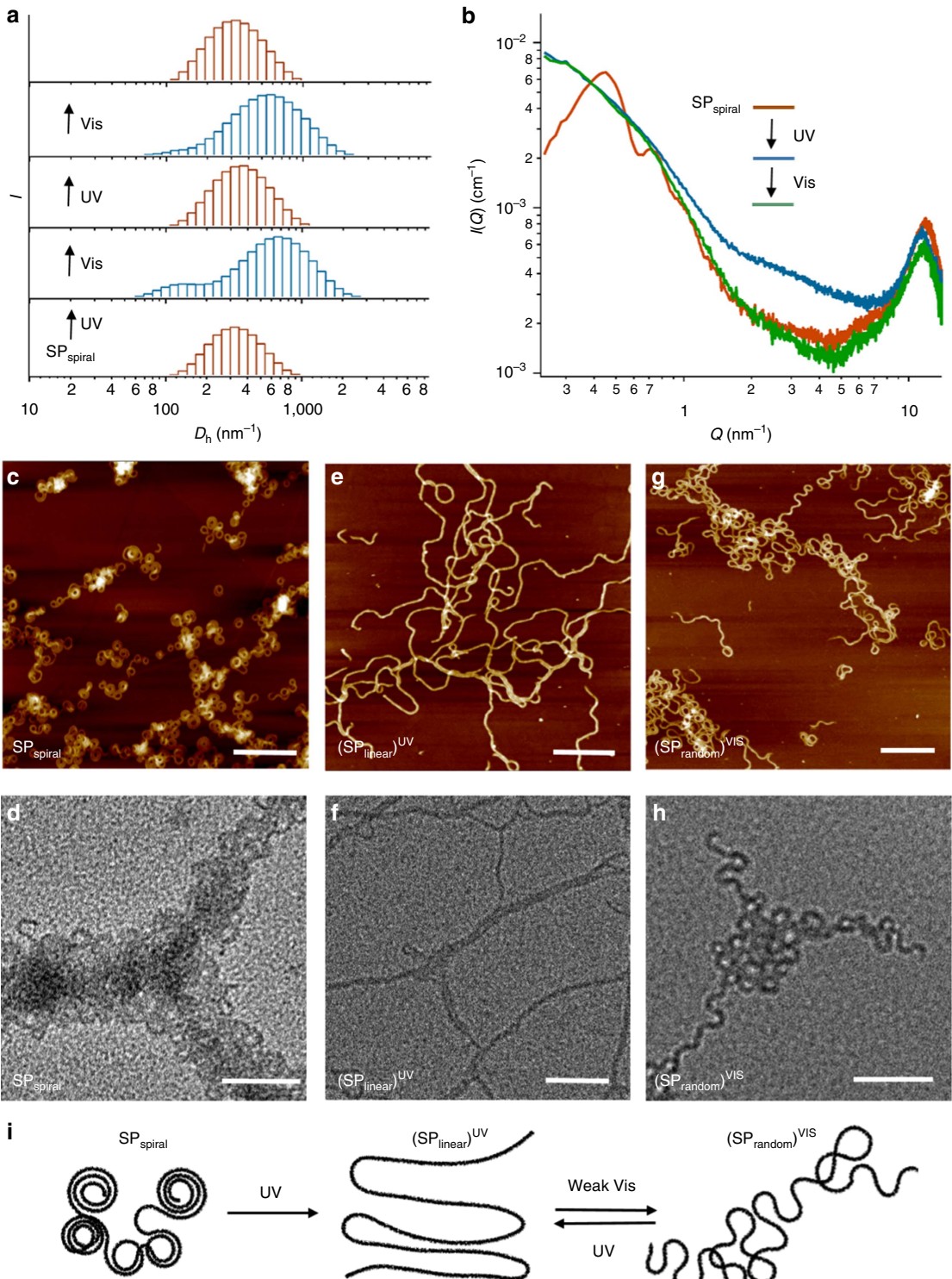

**Figure 4 | Morphological analysis of photochemically transmuted SPs.** (**a**) DLS analysis of SP$_{spiral}$ showing photoinduced changes in the distribution of the hydrodynamic diameters ($D_h$). SP$_{spiral}$ solutions ($c = 1.0 \times 10^{-4}$ M) were successively exposed to UV and Vis light; after each irradiation step, sample solutions were diluted to $c = 2.5 \times 10^{-5}$ M to reduce agglomeration. (**b**) Photoinduced change of the SAXS profiles of SP$_{spiral}$ ($c = 1.0 \times 10^{-4}$ M) upon successive exposure to UV and Vis light. (**c–h**) AFM (**c,e,g**; scale bar, 200 nm) and TEM images (**d,f,h**; scale bar, 100 nm) of SP$_{spiral}$ (**c,d**), (SP$_{linear}$)$^{UV}$ (**e,f**) and (SP$_{random}$)$^{Vis}$ (**g,h**). Samples were prepared by exposing SP$_{spiral}$ ($c = 1.0 \times 10^{-4}$ M) to UV irradiation for 20 min, followed by exposure to weak Vis light for 20 min. (**i**) Schematic representation of photochemically transmuted SPs. Irradiation with UV light transmutes spirally folded SP$_{spiral}$ into linear (SP$_{linear}$)$^{UV}$, which is further transmuted into randomly folded (SP$_{random}$)$^{Vis}$ by exposure to Vis light. (SP$_{linear}$)$^{UV}$ and (SP$_{random}$)$^{Vis}$ are interconvertible by light.

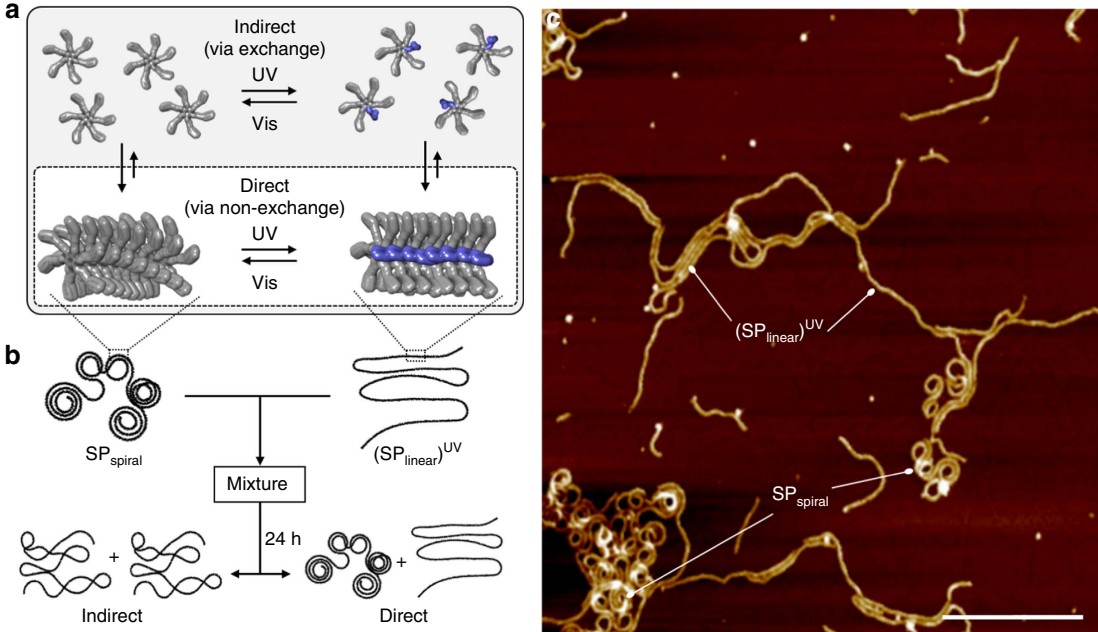

**Figure 5 | Mechanistic insight into photochemical transmutation of SPs.** (**a**) Schematic illustration of light-induced curvature changes of SP via two possible mechanisms. Grey rectangle: indirect mechanism, where isomerization of the azobenzene occurs in the individual molecular/rosette level via aggregate–monomer exchange. White rectangle: direct mechanism, where isomerization of the azobenzene moieties occurs within the stacked SP chains. (**b**) Schematic illustration of possible final morphologies in a mixing experiment for equimolar mixture of $SP_{spiral}$ and $(SP_{linear})^{UV}$. In the case where aggregate–monomer exchange occurs, a single morphology with mixed characters of the individual SPs will be observed. In the case where aggregate–monomer exchange does not occur or occurs very slowly, two individual SP morphologies will be retained after 24 h. (**c**) Experimental result of the mixing experiment of $SP_{spiral}$ and $(SP_{linear})^{UV}$ in MCH ($c = 1.0 \times 10^{-4}$ M for both). AFM analysis (scale bar, 200 nm) revealed that both SPs retained their individual morphological features in solution even after 1 day from the mixing.

hand, the isomerization of azobenzene that typically occur on subpicosecond to picosecond timescales[44] occur in the monomeric state given by aggregate–monomer exchange/reshuffling[45,46]. Although it is hard to testify the direct mechanism, we could rule out the indirect mechanism unequivocally by mixing separately prepared $SP_{spiral}$ and $(SP_{linear})^{UV}$ (Fig. 5b). When the mixture was kept for 24 h at 20 °C, no alteration in the individual structures was observed (Fig. 5c). This finding illustrates that the chain exchange/reshuffling in our SP system, if it occurs, is too slow to show appreciable morphology change in the given time scale. Hence, the loss and recovery of spontaneous curvature should occur directly through *in situ* photoisomerization of the azobenzene moieties in aggregated **2**. After irradiation with UV light, the 23% of *cis*-**2** content means that statistically one to two out of the six *trans*-**2** moieties per hexamer are converted into *cis*-**2**. Based on the stacking model of hexamerized **2**, the *trans*-azobenzene moieties, whose long axes are oriented nearly parallel to the long axis of the SP fibres, can isomerize (Fig. 2f). The isomerization of the azobenzene units should result in a linear alignment of the hexamers, which changes the internal order and reduces the spontaneous curvature, given that non-planar hexamers containing *cis*-**2** cannot stack with rotational displacement.

**Helically folded SP fibres with unfolding and refolding.** We wanted to determine if the photoinduced loss of curvature starts randomly at several points within a fibre, or if it occurs very homogeneously for every loops, or even if it starts at a single point and proceeds continuously from one terminus to the other. The random mechanism should afford linear alignments only locally, and such linear domains may not form a very straight fibre (Fig. 6a). This should result in a loss of neat internal order for a given fibre, although each linear alignment may possess high

levels of internal order. In case of the homogeneous mechanism, we should observe homogeneous changes in curvature for every loops simultaneously with the progress of isomerization (Fig. 6b). Conversely, the continuous mechanism should generate high levels of internal order throughout, which should result in the formation of very straight fibres (Fig. 6c). To address this issue unequivocally, we wanted to monitor the photoinduced morphology change for a SP with a more organized and homogeneous conformation, for example, a helical SP. The formation of increasingly organized structures as a function of decreasing cooling rates has already been discussed. Consequently, a very slow cooling rate (for example, 0.01 °C min$^{-1}$) may be able to afford more folded SPs, although this approach is experimentally impractical. Therefore, we changed the solvent system to a chloroform ($CHCl_3$)–MCH mixture, wherein the polar $CHCl_3$ should weaken the intermolecular interactions and guide the supramolecular polymerization towards more thermodynamic control. Eventually, we identified conditions (cooling rate: 0.1 °C min$^{-1}$; $c = 1.0 \times 10^{-4}$ M; $CHCl_3$:MCH = 15:85, v/v; Supplementary Discussion) that were able to afford helically folded $SP_{helical}$ (Fig. 6d; Supplementary Figs 13 and 14)[33,47–49]. While most $SP_{helical}$ are partially unfolded, their helical handedness is retained throughout the entire fibre to afford left- (*M*) and right-handed (*P*)-$SP_{helical}$ (Supplementary Fig. 14d,e), suggesting that a highly cooperative supramolecular polymerization occurred at a higher-order level. A SAXS analysis of an $SP_{helical}$ solution exhibited maxima/minima at $Q = 0.3$–1.2 nm$^{-1}$, arising from the spontaneous curvature ($r_{ave} = 10.0 \pm 0.2$ nm) of the helical loops (Supplementary Fig. 15).

The investigation of the unfolding process of $SP_{helical}$ confirmed the occurrence of a random isomerization of the azobenzene moieties, which reduces the overall internal order in the fibres. The AFM images (Fig. 6d–h; Supplementary Fig. 16)

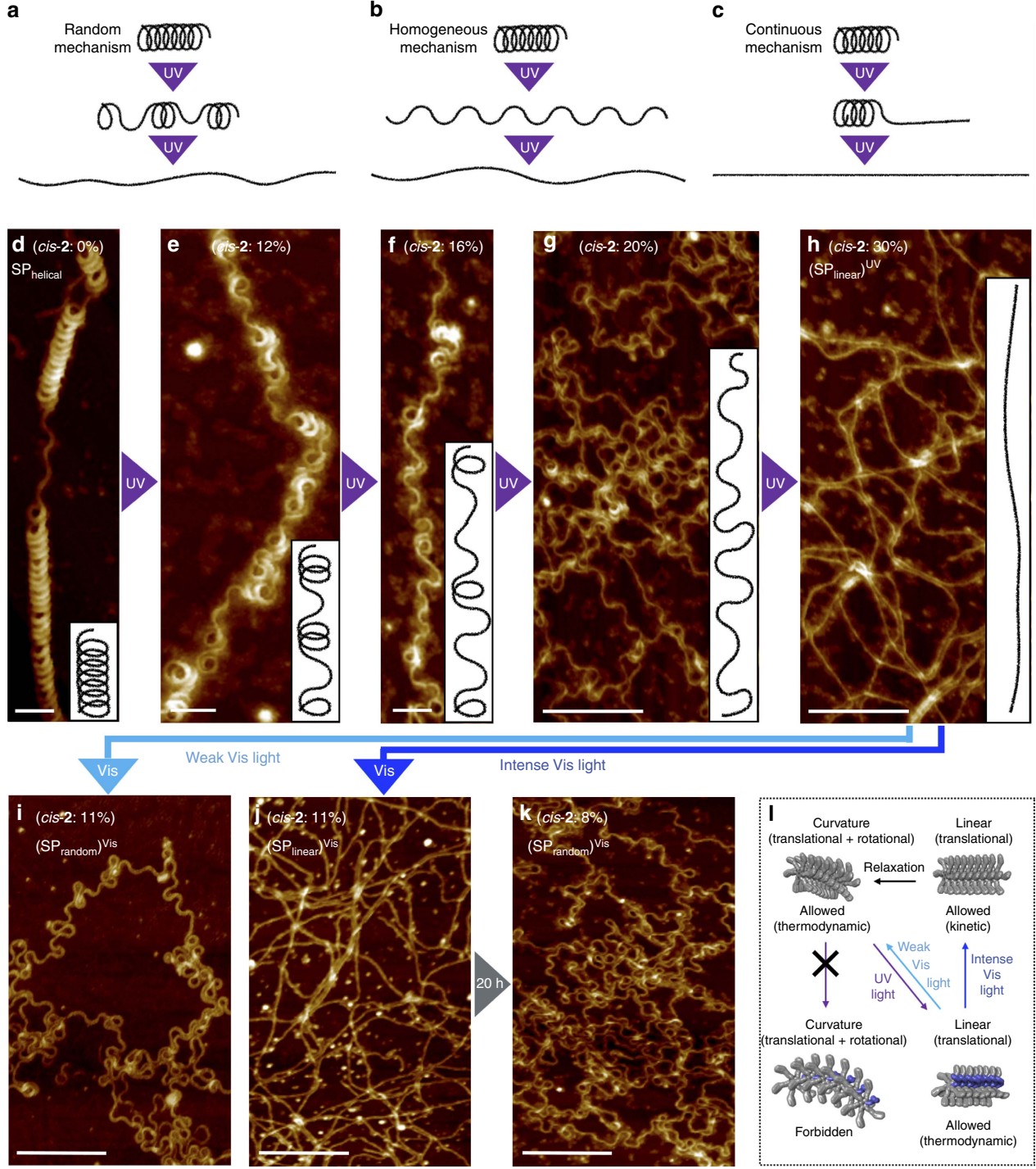

**Figure 6 | Light-induced unfolding of helical SP fibres and refolding of the resulting straight fibres into randomly coiled fibres.** (**a–c**) Schematic representation of three possible mechanisms for unfolding of SP$_{helical}$ with intermediate and final states. (**a**) Random mechanism, where photoinduced loss of curvature starts randomly at several points within a fibre. (**b**) Homogeneous mechanism, where loss of curvature occurs homogeneously at every loops simultaneously. (**c**) Continuous mechanism, where loss of curvature starts at one terminus and proceeds continuously towards the other terminus, leading to very straight fibre with high degree of internal order. (**d–h**) AFM images showing an unfolding process of SP$_{helical}$ into (SP$_{linear}$)$^{UV}$ upon gradually increasing the *cis*-**2** content by irradiation with UV light. (**i**) AFM image of (SP$_{random}$)$^{Vis}$ obtained from irradiation of (SP$_{linear}$)$^{UV}$ with weak Vis light for 20 min. Exposure to Vis light is able to steadily recover the spontaneous curvature. (**j**) AFM image of (SP$_{linear}$)$^{Vis}$ obtained from irradiation of (SP$_{linear}$)$^{UV}$ with intense Vis light for 30 s. Even though irradiation with Vis light causes a drop of *cis*-**2** content from 30 to 11%, the spontaneous curvature does not recover immediately to produce straight fibres (kinetic product). (**k**) AFM image of (SP$_{random}$)$^{Vis}$, which develops spontaneously and slowly from (SP$_{linear}$)$^{Vis}$ over 20 h. Scale bar, 100 nm (**d–f**) or 200 nm (**g–k**). (**l**) Schematic illustration of the loss and recovery of spontaneous curvature upon exposure to UV and Vis light, respectively. The curved stacks transform directly into linear stacks via *trans*-to-*cis* isomerization of the azobenzene moieties, as the non-planar hexamers involving *cis*-**2** cannot stack with rotational offset due to the steric repulsion arising from the *cis*-azobenzene moieties. For hexamers involving *cis*-**2**, only linear stacking (translational displacement) is possible. The recovery of the spontaneous curvature by Vis light depends on the intensity of the light source.

show the morphology changes of SP$_{helical}$ as a function of the content of cis-**2**, which is controlled by irradiation with UV light. The unfolding occurs in a highly controlled and linear fashion upon increasing the content of cis-**2**. At 12% cis-**2**, unfolded domains occur locally (Fig. 6e), and further increase up to 16% cis-**2** produces (SP$_{random}$)$^{UV}$ (Fig. 6f). At 20% cis-**2**, most helical domains disappear to generate waving fibres ($\theta_{ave} < 180°$) (Fig. 6g). In the presence of CHCl$_3$, the content of cis-**2** can be further increased, and a drastic change in curvature occurs up to 30% cis-**2**, where most of the curvature is lost to afford more extended (SP$_{linear}$)$^{UV}$ fibres (Fig. 6h). However, (SP$_{linear}$)$^{UV}$ fibres are not very straight (Fig. 6h), again supporting the random isomerization occurs in this case. As a proof of principle, cis-**2** monomers (in fact a mixture of trans-**2**:cis-**2** = 60:40) were subjected to thermal polymerization conditions, which afforded very straight fibres with long-range domains ($\sim 500$ nm) as a result of cooperatively stacking of well-distributed cis containing rosettes by aligning all cis-arms linearly (Supplementary Figs 17 and 18).

With these highly extended (SP$_{linear}$)$^{UV}$ fibres in hand, we examined if the recovery of spontaneous curvature occurs linearly with gradual decrease in the content of cis-**2**. When (SP$_{linear}$)$^{UV}$ was exposed to weak Vis light, while progressively diminishing the cis-**2** content (Supplementary Fig. 19), the spontaneous curvature gradually recovered to produce (SP$_{random}$)$^{Vis}$ (Fig. 6i; Supplementary Fig. 20). Then, we wanted to find out if an abrupt reduction of the cis-**2** content in the planar trans-**2** hexamers would lead to an immediate formation of their thermodynamically preferred stacking mode (spontaneous curvature) or if the kinetically preferred stacking mode (linear stacks) would be maintained for some time (Fig. 6l). Interestingly, an exposure of (SP$_{linear}$)$^{UV}$ to intense Vis light did not induce any apparent morphology change (Fig. 6j). This result illustrates that the photochemical formation of kinetic (SP$_{linear}$)$^{Vis}$ fibres lacks the high degree of internal order that is required to generate spontaneous curvature, observed for thermally obtained SP$_{linear}$. (SP$_{linear}$)$^{Vis}$ slowly converts into (SP$_{random}$)$^{Vis}$ over the course of 20 h (Fig. 6k; Supplementary Fig. 21), which is similar to the kinetic product SP$_{linear}$.

## Discussion

In conclusion, this study demonstrates that a SP approach may confer 1D fibres that exhibit levels of higher-order conformations similar to those of polypeptides. The strong advantage of this strategy is that these higher-order conformations can be controlled by modulating the internal staking order by non-invasive external stimuli such as light. The observed nonlinear morphological change of photochemically regulated fibres upon exposure to intense Vis light could be useful for the synthesis of supramolecular nanomaterials that may be able to store photon energy in kinetically favoured structures. The remote control over the internal order of non-equilibrated/robust SP fibres to execute conformational changes may advance the frontiers of SP towards applications by mimicking the functionality of biopolymer. Although the photoinduced disassembly–reassembly of 1D aggregates[50,51] has already been realized, the light-induced unfolding–refolding approach presented herein guides supramolecular as well as polymer chemistry towards unchartered territory. However, in the present study, the unfolding–refolding of helical coils is not fully reversible with respect to the first refolding. Achieving complete light-controlled reversibility between helical coils and unfolded structures represents the next goal of our research. This may be accomplished by inducing complete and continuous back-isomerization of azobenzene from one termini to the other of the non-equilibrated SP fibres.

## Methods

**Materials.** Compound **2** was synthesized according to Supplementary Fig. 23. As for characterization, $^1$H NMR, $^{13}$C NMR and HRMS of compound **2** were included in Supplementary Methods. Commercially available reagents and solvents were of reagent grade and used without further purification. MCH and CHCl$_3$, the solvents used as media for supramolecular polymerization, were all spectral grade and used without further purification.

**Small-angle X-ray scattering measurements.** SAXS measurements were carried out at BL-6A at the Photon Factory of the High Energy Accelerator Research Organization (KEK) in Tsukuba, Japan[52,53]. The sample solutions were placed in the stainless-steel sample cell with the light path length of 1 mm and 20 μm-thick quartz glass windows. The cell was maintained at around 293 K. PILATUS3 1 M (DECTRIS) was used as a detector. The X-ray wavelength was adjusted to 1.5 Å, and the sample-detector distance was 481 and 1966 mm, calibrated using a silver behenate as a standard sample. These settings provided a detectable $Q$-range of order 0.2–15 and 0.05–2.5 nm$^{-1}$, respectively. Sixty frames were collected with the exposure time of 10 s. Since any radiation damages were not seen in all data, all the data in each sample were averaged to improve signal-to-noise ratio. The total integration time was 600 s. The two-dimensional scattering data were circularly averaged to convert into the 1D scattering intensity data, normalizing the scattering intensity to the absolute intensity scale (cm$^{-1}$) by using water as a reference. The background subtraction was also performed to get the final scattering intensities. The magnitude of the scattering vector is given by $Q = (4\pi/\lambda)\sin\theta/2$, where $\lambda$ is the X-ray wavelength and $\theta$ is the scattering angle. The software SAngler was used for these data processing[54].

**Photo-irradiation experiments of SPs.** For the trans-to-cis isomerization of trans-**2** in SP$_{spiral}$, a UV LED lamp ($\lambda = 365$ nm) with the intensity of 17 W cm$^{-2}$ (at a distance of 5 cm and using its maximum intensity) was used as a light source. The lamp was placed at a distance of 1 cm from a quartz cuvette (path length: 1 mm) containing SP solution and the UV light was irradiated. Under this condition, an apparent photostationary state (PSS) is typically achieved within 20 min of UV exposure. The photoinduced unfolding of SP$_{helical}$ into (SP$_{linear}$)$^{UV}$ was performed under the careful control of increasing content of cis-**2** by the LED lamp with increasing irradiation time and regulating the light intensity upon changing the distance between the cuvette and the light source. When solutions of SPs consisting of trans-**2** were exposed to the UV light, a decrease in absorbance around 370 nm was observed in UV–Vis spectra, indicating the extent of photoisomerization of azobenzene unit. The content of cis-**2** in UV-irradiated SPs in MCH was determined by comparing the decrease in absorbance intensity of the absorption maximum of the trans-**2** by referring the absorption change of trans-**2** upon UV irradiation in CDCl$_3$, wherein the content of the cis-isomer was determined by $^1$H NMR spectroscopy. For the slow cis-to-trans back-isomerization of (SP$_{linear}$)$^{UV}$, a Vis LED lamp ($\lambda = 470$ nm) at its minimum intensity was used as a light source. The lamp was placed at a distance of 20 cm from a cuvette (path length: 1 mm) containing SP solution and the Vis light was irradiated. Under this condition, about 20 min irradiation is required to reach a PSS. For the fast cis-to-trans isomerization using stronger Vis light, two Vis LED lamps were placed at a distance of 0.5 cm from two opposite sides of the cuvette (path length: 1 mm) containing SP solution. Under this condition, about 30-s irradiation is enough to reach the PSS where the same content of cis-**2** is achieved in comparison with the above condition.

**Data availability.** All data supporting the findings of this study are available within the article and its Supplementary Information files, and from the corresponding author upon reasonable request.

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

## Acknowledgements

This work was supported by KAKENHI (no. 26102010), a Grant-in-Aid for Scientific Research on Innovative Areas 'π-Figuration' (no. 26102001) of The Ministry of Education, Culture, Sports, Science and Technology, Japan. B.A. thanks the International Research Fellowship of the Japan Society for the Promotion of Science. This work was performed under the approval of the Photon Factory Program Advisory Committee (Proposal No. 2015PF-22, 2015PF-29 and 2016PF-04).

## Author contributions

S.Y. and B.A. conceptualized the project. B.A. have performed most of the experiments described in the manuscript. Y.Y. synthesized the molecule **2** and did some initial experiments. N.S., H.T., R.H. and S.-i.A. collected SAXS data. M.J.H. simulated the SAXS data and wrote SAXS section of the manuscript. T.O. performed the TEM experiments. B.A. and S.Y. wrote the overall manuscript. S.Y., B.A. and M.Y. worked on the figures. All authors including M.Y., K.W., X.L., K.A., T.O. and T.K. discussed the results and commented on the manuscript. The overall project management was by S.Y.

## Additional information

**Competing interests:** The authors declare no competing financial interests.

