## [Peer Review File · Nature Communications]

Reviewers' comments:

Reviewer #1 (Remarks to the Author):

The authors disclose a very intriguing study to the folding and unfolding of a new set of supramolecular polymers. Although the photo-switching of (organo)gels (Feringa) and the light-induced polymerization-depolymerization of porphyrins (Meijer) has been published, the approach presented here is new and without doubt of great interest to everyone in the field of supramolecular and polymer chemistry.

The manuscript beautifully shows that Supramolecular Polymers are indeed a great new contribution to polymer science. The results are very convincing. The detailed studies with a great series of techniques makes the work very comprehensive. All aspects are presented in a clear way and no errors or missing elements can be found; the authors are congratulated with this very nice piece of work. One point needs to be addressed and that is related to the (ir)reversibility of the folding after first refolding. Where the reader gets the idea from the title, intro and most of the text that it is fully reversible, the last part of the manuscript shows that the refolding is a bit more complicated. Why is the system not fully reversible? What hampers the back folding? And is the pathway complexity even more complex than know today? In the revised version, the authors can probably also cite the work mentioned above.

Reviewer #2 (Remarks to the Author):

First of all, congratulations to the authors for this very innovative piece of work! They have come up with a clever design, which was subsequently thoroughly executed including lots of impressive characterization data, in particular AFM but also SAXS and DLS. The work is novel and of very high quality, so I am very much in favor of publication of this excellent work after few improvements have been implemented:

Irradiation experiments (Figure 4): Why do the authors irradiate a sample of higher concentration (1.0×10^{-4} M) and subsequently dilute (to 2.5×10^{-5} M)? They claim that this necessary to reduce agglomeration but how do they distinguish between agglomeration of self-assembled chains and the self-assembly of the chain itself? Typically, irradiation experiments are limited by optical density, i.e. preventing the use of solutions of higher concentration as light cannot penetrate and irradiations take a long time, causing damage of the sample. In this case, obviously it's the other way around and hence the should also carry out direct irradiation of the 2.5×10^{-5} M solutions!

Assembly of cis-enriched 2 (Figure): The authors claim his case they observe formation of long linear chains because the cis-monomers assemble into cis-only rosettes, which then can be integrated into the fiber next to trans-only rosettes. While I see such process of self-sorting to be very likely operating - it is a common form of segregation on the molecular (assembly) level - I think that this should give rise to a less uniform fiber structure, i.e. the diameter should vary. The authors should analyze their data to see if this indeed is the case. Furthermore, they should provide evidence for their claim on page 11: "result of cooperatively stacking of well-distributed cis containing rosettes by aligning all cis-arms linearly". This should involve also a molecular model of the all-cis rosette.

Wording: The authors should differentiate between supra- or intermolecular, non-covalent and macro- or intramolecular, covalent structures. Therefore, the words "conformation" as well as "folding" and "unfolding" should only be used in an intramolecular context but not when referring to supramolecular fibers. Please change the text, in particular the title as well as the abstract, accordingly.

Minor spelling mistakes: Correct "azobenzene" in page 11 and "methylcyclohexane" in the caption of Figure 3.

Reviewer #3 (Remarks to the Author):

In this manuscript, Yagai and Co-workers report a unique study, wherein light-stimuli has been used to manipulate the conformation of a supramolecular polymer. Although conformational control of covalent polymeric analogues have been studied extensively for the design of nanoparticles, to the best of my knowledge this is the first time a similar attempt has been made in a supramolecular polymer. The studies have been carried out in a systematic way with detailed mechanistic investigations and various experimental techniques. I strongly recommend the acceptance of this article in Nature Communications.

However, I would like the author to address following comments:

1. Authors mention on Pg 5 line 108, that introduction of trans-azobenzene does not affect the internal order. While as per experiments morphology of "trans-2" is distinctly different from "1", providing one of the many proofs that "trans-2" does have an effect on internal order. Furthermore, it is perhaps possible to have similar internal curvature (which is what authors are alluding to) with changes in internal order. Therefore in my opinion, "trans-2" changes the internal order but maintains the internal curvature. Thus the above mentioned line should be suitably edited.
2. Authors use the UV absorption region between 410-430 nm as a marker for internal order. This notion perhaps derives from changes in figure 2g where monomer to aggregate changes happen in this region. This however is accompanied by changes in 450-500 nm region, which is not evident. Furthermore one must realise that changes in this region are also susceptible to isomerization changes (SI figure 7). Thus concluding internal order from SI figure 6 does not seem suitable. As a measure of internal order one can look at the heating curves of the three forms (random coil, linear, spiral) and see if the T_e in these cases follows a trend such that spiral > random coil > linear.
3. On Pg 7 line 159: The reversibility of the DLS changes in figure 4a has been highlighted. One can however notice that DLS after the UV-vis cycles is slightly larger than the initial state. Though this change is less, but goes to show that obtained structures after cycles are random coils and not complete spiral (thus larger D_h than the initial spiral). Sentence can thus be suitably edited.
4. DLS has been done with further dilution to avoid cross-linked structures. How good is the kinetic stability of the individual supramolecular chains to ensure that this dilution does not affect the analyses! Does the Number % in DLS analyses also give a similar trend?
5. I am very curious to see whether mixed rosettes can be made with five molecules of 1 and one molecules of 2. Do they self-sort or co-assemble. If they co-assemble it would be interesting to see a similar conformational control! However this study is certainly not required for the present manuscript.

Answer to the comments raised by referee 1 (Additions to the main text are highlighted yellow)

General Comments (1):

The authors disclose a very intriguing study to the folding and unfolding of a new set of supramolecular polymers. Although the photo-switching of (organo)gels (Feringa) and the light-induced polymerization-depolymerization of porphyrins (Meijer) has been published, the approach presented here is new and without doubt of great interest to everyone in the field of supramolecular and polymer chemistry.

Author's Response:

We would like to thank the reviewer and we are delighted about his/her comment that “the approach presented here is new and without doubt of great interest to everyone in the field of supramolecular and polymer chemistry.”.

General Comments (2):

The manuscript beautifully shows that Supramolecular Polymers are indeed a great new contribution to polymer science. The results are very convincing. The detailed studies with a great series of techniques makes the work very comprehensive. All aspects are presented in a clear way and no errors or missing elements can be found; the authors are congratulated with this very nice piece of work.

Author's Response:

Thank you very much for this kind and generous comment. We appreciate the reviewer's in-depth understanding of our manuscript and the highly specialized comments.

Specific Comments (1):

One point needs to be addressed and that is related to the (ir)reversibility of the folding after first refolding. Where the reader gets the idea from the title, intro and most of the text that it is fully reversible, the last part of the manuscript shows that the refolding is a bit more complicated. (i) Why is the system not fully reversible? (ii) What hampers the back folding? (iii) And is the pathway complexity even more complex than know today?

Author's Response:

Please allow us to address these issues separating into three parts (i)–(iii).

(i) The incomplete reversibility of the first cycle is explained by the following short paragraph, which can be found in the section “Photochemically transmuted SPs” in the original manuscript. “The recovery of the curvature induced by Vis light could simultaneously occur locally throughout the polymer backbone without maintaining a specific turning direction, thus resulting in randomly folded coils with a lower θ_{ave} . Hence, sufficient preservation of the turning

direction at each curve was achieved with a higher θ_{ave} by thermal supramolecular polymerization of the monomers.”

However, in order to avoid any potential confusion of the reader with respect to the (ir)reversibility of the folding, we included the following paragraph into the “Discussion” section of the revised manuscript.

“However, in the present study, the unfolding-refolding of helical coils is not fully reversible with respect to the first refolding. Achieving complete light-controlled reversibility between helical coils and unfolded structures represents the next goal of our research. This may be accomplished by inducing complete and continuous back-isomerization of azobenzene from one termini to the other of the non-equilibrated SP fibers.”

(ii) The random back-isomerization of azobenzene moieties hampers complete refolding of the fibers into spiral or helical structures. This random back-isomerization-induced recovery of curvature cannot maintain a specific turning direction throughout long fibers, which results in the formation of \mathbf{SP}_{random} . Moreover, the incomplete back-isomerization may be partially responsible for the incomplete refolding. For instance, irradiation of $(\mathbf{SP}_{linear})^{UV}$ with Vis light recovered *cis-2* only up to 11%, while a subsequent thermal back-isomerization at room temperature recovered *cis-2* up to 5% after 60 h. For 11% and 5% *cis-2*, θ_{ave} values of 267° and 335° were observed, respectively. Our SAXS data in the original manuscript revealed that $(\mathbf{SP}_{random})^{Vis}$ with 11% *cis-2* was unable to regain the specific scattering derived from the spontaneous curvature due to the lower θ_{ave} . After submission of the paper, we conducted further SAXS experiments on $(\mathbf{SP}_{random})^{Vis}$ with 5% *cis-2*. This new measurements showed weak contributions from the specific scattering derived from the spontaneous curvature, thus supporting an improved reversibility of the foldability. Consequently, incomplete back-isomerization induced by Vis light should also be considered partially responsible for the incomplete refolding. For the benefit of the reader, the additional SAXS experiments were included in the revised Supplementary Information as Supplementary Figure 9.

(iii) We agree with the reviewer that the involvement of photoisomerization processes should render existing pathways in the present system more complex than usual due to the far-from-equilibrated nature. Given a sufficiently fast rate of monomer-aggregate exchange under the experimental conditions, the formation of the thermodynamically stable $\mathbf{SP}_{helical}$ should be expected after back-isomerization under suitable conditions. However, $\mathbf{SP}_{helical}$ is not obtained upon back-isomerization, due to the very slow (or even absent) monomer-aggregate exchange. Therefore, some supramolecular structures could only be obtained via the pathways

that involve photoisomerization processes. Non-equilibrated photoresponsive systems may thus open a new avenue in the area of supramolecular polymerizations, especially with respect to product diversification, which is currently under investigation in our laboratory.

Specific Comment (2):

Although the photo-switching of (organo)gels (Feringa) and the light-induced polymerization-depolymerization of porphyrins (Meijer) has been published, the approach presented here is new and without doubt of great interest to everyone in the field of supramolecular and polymer chemistry. In the revised version, the authors can probably also cite the work mentioned above.

Author's Response:

We agree with this suggestion, and added the following two references to the revised manuscript.

50. de Jong, J. J. D. Lucas, L. N. Kellogg, R. M. van Esch & J. H. Feringa, B. L. Reversible optical transcription of supramolecular chirality into molecular chirality. *Science* **304**, 278–281 (2004).

51. Hirose, T. Helmich, F. & Meijer, E. W. Photocontrol over cooperative porphyrin self-assembly with phenylazopyridine ligands. *Angew. Chem. Int. Ed.* **52**, 304–309 (2013).

Moreover, the following sentence was added to the “Discussion” section of the revised manuscript:

"Although the photo-induced disassembly-reassembly of one-dimensional aggregates^{50,51} has already been realized, the light-induced unfolding-refolding approach presented herein is unprecedented and guides supramolecular as well as polymer chemistry toward uncharted territory."

Answer to the comments raised by referee 2 (Additions to the main text are highlighted green)

General Comments

First of all, congratulations to the authors for this very innovative piece of work! They have come up with a clever design, which was subsequently thoroughly executed including lots of impressive characterization data, in particular AFM but also SAXS and DLS. The work is novel and of very high quality, so I am very much in favor of publication of this excellent work after few improvements have been implemented:

Author's Response:

Thank you very much for the kind comments/remarks and the in-depth understanding of our manuscript. All the comments raised by this referees were addressed in the revised manuscript.

Specific Comments (1):

Irradiation experiments (Figure 4): Why do the authors irradiate a sample of higher concentration (1.0×10^{-4} M) and subsequently dilute (to 2.5×10^{-5} M)? They claim that this necessary to reduce agglomeration but how do they distinguish between agglomeration of self-assembled chains and the self-assembly of the chain itself? Typically, irradiation experiments are limited by optical density, i.e. preventing the use of solutions of higher concentration as light cannot penetrate and irradiations take a long time, causing damage of the sample. In this case, obviously it's the other way around and hence the should also carry out direct irradiation of the 2.5×10^{-5} M solutions!

Author's Response:

We initially set the concentration of the AFM studies to 1.0×10^{-4} M, in order to increase the population/visibility of analyzable supramolecular fibers in an image of a specific dimensions. We also carried out DLS experiments upon photoirradiation at 1.0×10^{-4} M, but were unable to observe any significant changes. We thought that this should be due to agglomeration of fibers, as the AFM data showed crowding of fibers. Subsequently, we carried out DLS experiments after lowering the concentration to 2.5×10^{-5} M, and obtained results that agree well with the unfolding/refolding of individual fibers that was observed by AFM. We were also able to confirm by AFM that this dilution does not affect the length of the fibers, i.e., the degree of polymerization, which is most likely due to the strong non-equilibrated nature of the assemblies. However, following the suggestion of the reviewer, we also carried out a DLS experiment upon direct irradiation of the diluted solution (2.5×10^{-5} M). The results were almost identical to those for the concentrated solution (1.0×10^{-4} M), which was photoirradiated prior to the dilution to 2.5×10^{-5} M for DLS measurement. This result was added as “**Supplementary Figure 8**” to the revised version of the Supplementary Information.

Specific Comments (2):

Assembly of cis-enriched **2** (Figure): (i) The authors claim in this case they observe formation of long linear chains because the cis-monomers assemble into cis-only rosettes, which then can be integrated into the fiber next to trans-only rosettes. While I see such process of self-sorting to be very likely operating - it is a common form of segregation on the molecular (assembly) level - I think that this should give rise to a less uniform fiber structure, i.e. the diameter should vary. The authors should analyze their data to see if this indeed is the case. (ii) Furthermore, they should provide evidence for their claim on page 11: "result of cooperatively stacking of well-distributed cis containing rosettes by aligning all cis-arms linearly". This should involve also a molecular model of the all-cis rosette.

Author's Response:

Please allow us to answer these comments separately:

(i) Perhaps the referee misunderstood the point we were trying to make here, as we did not claim that the formation of long linear SP chains (~ 500 nm domain) is due to the self-sorting at the rosette level, where *cis*-monomers assemble into *cis*-only rosettes, which can subsequently be integrated into the fiber next to *trans*-only rosettes. In the original manuscript, we merely proposed the formation of heteromeric rosettes (in which *trans-2* and *cis-2* co-aggregate in a well-distributed fashion to form each rosette). The ensuing stacking of these heteromeric rosettes affords very straight fibres by linearly aligning *cis*-arms, which is favorable according to the molecular modeling results, as the formation of SPs occurs upon stacking of individual rosettes under thermal supramolecular polymerization conditions. For the stacking of self-sorted rosettes, individual fibers with non-uniform diameters should be observed (as suggested by the referee). However, we always obtain fibres with a homogenous diameter for SPs with *cis*-enriched **2**, which excludes the possibility of stacking *trans*-only and *cis*-only rosettes.

(ii) The claim "...result of cooperatively stacking of well-distributed *cis*-containing rosettes by aligning all *cis*-arms linearly" is mainly based on the results of AFM measurements and molecular modeling. To obtain information at the molecular level, we analyzed thin films of our supramolecular fibers by powder X-ray diffraction using synchrotron radiation. However, the thin films prepared from $\text{SP}_{\text{spiral}}$ and $(\text{SP}_{\text{linear}})^{\text{UV}}$ showed no significant difference, although sharp diffraction corresponding to the fiber width was observed for both samples. A further elucidation of the packing structure may require aligned fibre samples, which is beyond the scope of this study. In response to the referee's request, a schematic model for stacking of heteromeric rosettes was added as Supplementary Figure 18.

In addition to the results from molecular modeling, our proposed model "cooperatively stacking of well-distributed *cis*-containing rosettes by aligning all *cis*-arms linearly" is also based on the following results. Fibers composed of *cis*-containing heteromeric rosettes are

supported by the sole formation of straight fibers. Moreover, we have already ruled out the possibility of mixed stacking for self-sorted rosettes. The formation of heteromeric rosettes with well-distributed *trans-2* and *cis-2* at the rosette level is expected upon polymerization by cooling following UV irradiation at 105 °C. Due to the steric demand of the *cis*-arms, the stacking of these heteromeric rosettes should occur by accommodating *cis*-arms in the space generated by the isomerization, which should afford very straight fibers with long-range domains (~500 nm).

Specific Comments (3):

Wording: The authors should differentiate between supra- or intermolecular, non-covalent and macro- or intramolecular, covalent structures. Therefore, the words "conformation" as well as "folding" and "unfolding" should only be used in an intramolecular context but not when referring to supramolecular fibers. Please change the text, in particular the title as well as the abstract, accordingly.

Author's Response:

We appreciate the referee's reservations, but in the present study, we would like to use the terms "conformation" as well as "folding" and "unfolding" for our supramolecular polymers due to the following reasons:

This terminology has been used for the helical supramolecular polymerization of tobacco mosaic virus (TMV) capsid proteins, wherein intermolecular interactions are responsible for the formation of helical tubular structures. These terms have recently also been used in the field of synthetic supramolecular polymer to describe similar phenomena. As an example, we would like to show a sentence from a review article of M. Lee (Journal of Polymer Science: Part A: Polymer Chemistry):

“In this supramolecular polymer, each of the two merocyanine units in one monomer dimerize with another dye in an antiparallel fashion, thereby resulting in the formation of polymeric chains that are folded into helical conformations in a nonpolar solvent.”

Apart from this example, there are numerous other reports on “helical conformations” in the field of supramolecular chemistry. The use of the folding/unfolding terminology is still not common in the context of supramolecular polymers, presumably due to the fact that similar phenomena have not been commonly realized yet, i.e., external stimuli usually dissociate the main chain of supramolecular polymer into monomers. However, in this study, the unfolding process of our robust supramolecular polymer chains does not involve monomer assembly/disassembly. The observed phenomena are very similar to those of covalent/biopolymer counterparts. Therefore, we would like to extend this terminology to supramolecular polymers. However, we would also gratefully welcome any suggestions from the reviewer regarding an alternative terminology that fits well with our observations.

Specific Comments (4):

Minor spelling mistakes: Correct "azobenzene" in page 11 (authors: this might be page 9) and "methylocyclohexane" in the caption of Figure 3.

Author's Response:

We would like to thank the referee for his/her careful observations. These spelling mistakes were corrected.

Answer to the comments raised by referee 3 (Additions to the main text are highlighted pink)

General Comments

In this manuscript, Yagai and Co-workers report a unique study, wherein light-stimuli has been used to manipulate the conformation of a supramolecular polymer. Although conformational control of covalent polymeric analogues have been studied extensively for the design of nanoparticles, to the best of my knowledge this is the first time a similar attempt has been made in a supramolecular polymer. The studies have been carried out in a systematic way with detailed mechanistic investigations and various experimental techniques. I strongly recommend the acceptance of this article in Nature Communications. However, I would like the author to address following comments:

Author's Response:

We would like to thank the reviewer for his/her in-depth understanding of our study and the insightful comments.

Specific Comments (1)

Authors mention on Pg 5 line 108, that introduction of trans-azobenzene does not affect the internal order. While as per experiments morphology of *trans-2* is distinctly different from **1**, providing one of the many proofs that *trans-2* does have an effect on internal order. Furthermore, it is perhaps possible to have similar internal curvature (which is what authors are alluding to) with changes in internal order. Therefore in my opinion, *trans-2* changes the internal order but maintains the internal curvature. Thus the above mentioned line should be suitably edited.

Author's Response:

Actually the internal curvature (suggested by the referee) changes slightly, as the r_{ave} value for a loop of **SP_{random}** (10.4 ± 0.2 nm) is higher than that of a nanoring of **1** (7 ± 0.1 nm). However, the point we tried to make here is that the "degrees of internal order" are responsible for generating the spontaneous curvature, and that these do not change upon introduction of azobenzene moieties. A comparable degree of internal order is required to afford similar spontaneous curvature in a nanoring of **1** or in a loop in **SP_{random}** of *trans-2*. Of course, different nanostructures/morphologies should exhibit different types of internal order. As **SP_{random}** could be formed by mere extension of the nanorings upon further supramolecular polymerization in the presence of stronger cohesive forces, we are convinced that the degrees of internal order, which are accountable for the spontaneous curvature, are almost identical.

In agreement with the referee and in order to eliminate any potential ambiguity, we revised the corresponding part of the manuscript from

Original: “The similarity in the appearance of the spontaneous curvature of $\text{SP}_{\text{random}}$ ($r_{\text{ave}} = 10.4 \pm 0.2$ nm, $\theta_{\text{ave}} = 370 \pm 5^\circ$) (Fig. 3f,g) with that of the toroids of **1** indicates that the introduction of a *trans*-azobenzene unit in **2** does not affect the internal order.”

to

Revised: “The similarity in the appearance of the spontaneous curvature of $\text{SP}_{\text{random}}$ ($r_{\text{ave}} = 10.4 \pm 0.2$ nm, $\theta_{\text{ave}} = 370 \pm 5^\circ$) (Fig. 3f,g) with that of the toroids of **1** indicates that the introduction of a *trans*-azobenzene unit in **2** does not considerably affect the internal order responsible for the curvature.”

in the revised version.

Specific Comments (2):

(i) Authors use the UV absorption region between 410-430 nm as a marker for internal order. This notion perhaps derives from changes in figure 2g where monomer to aggregate changes happen in this region. This however is accompanied by changes in 450-500 nm region, which is not evident. (ii) Furthermore one must realise that changes in this region are also susceptible to isomerization changes (SI figure 7). Thus concluding internal order from SI figure 6 does not seem suitable. (iii) As a measure of internal order one can look at the heating curves of the three forms (random coil, linear, spiral) and see if the T_e in these cases follows a trend such that spiral > random coil > linear.

Author’s Response:

Please allow us to address these comments separately:

(i) We partially agree with the referee that changes in the 450–500 nm region (J band of naphthalene) should also be expected. However, given the broad features of the J-band and considering that the absorption changes of the sharp band in the region 410–430 nm are small, significant changes in this region are not likely. The thorough investigation of the thermal and photo-induced transmutations of SPs in the present study allowed us to conclude that the absorption in the 410-430 nm region may be used as a marker to measure subtle differences of internal order, as the absorption intensities are commensurate with the visual internal order observed by AFM. However, we are unable to determine with confidence whether such small changes in the 410–430 nm region should be attributed to the monomeric nature of the naphthalene moieties or to other factors.

(ii) Based on our extensive experimental data, we are relatively confident that the 410–430 nm region reflects only the naphthalene moieties. The changes in the absorption that are shown in

Supplementary Figure 6 should not necessarily be expected to correlate with Supplementary Figure 7, as the isomerization occurs in different solvents (CHCl_3 and MCH). Moreover, a decrease in absorbance in the 410–430 nm region has to be considered for the photoisomerization in CHCl_3 . For the photoisomerization in MCH (Supplementary Figure 6), the absorbance increases in the 410–430 nm region, which is not due to the isomerization, as evident from a cross-check with reference compound **6** (on page 7 of the Supplementary Information) and **1** (no azobenzene moieties) in MCH. Accordingly, we think that the 410–430 nm region shows the internal order arising from subtle differences in naphthalene stacking. Similar trends were observed in Supplementary Figure 3 (dependence on the cooling rate) and Supplementary Figure 6 (dependence on the isomerization), which corroborate the morphologies.

(iii) Prior to submitting the manuscript, we repeatedly measured the variable-temperature UV/vis spectra of different SPs upon slow heating. However, we did not see any significant differences in the resulting heating curves, albeit that they show thermal hysteresis around the critical temperature (unfortunately, an in-depth discussion of this point is beyond the scope of the current study, although we are currently exploring this feature using other molecules). The results, however, suggest that the T_c values do not necessarily depend either on the internal order or on the secondary conformation of the SPs.

Review-only-Figure 1. Heating curves for $\text{SP}_{\text{spiral}}$ (red line) and $\text{SP}_{\text{linear}}$ (cyan line) in addition to the cooling curve for the generation of $\text{SP}_{\text{spiral}}$ (black line); heating rate: 0.1 C/min; solvent: MCH; $c = 2.5 \times 10^{-5}$ M; path length of the cuvette: 10 mm.

Specific Comments (3):

On Pg 7 line 159: The reversibility of the DLS changes in figure 4a has been highlighted. One

can however notice that DLS after the UV-vis cycles is slightly larger than the initial state. Though this change is less, but goes to show that obtained structures after cycles are random coils and not complete spiral (thus larger D_h than the initial spiral). Sentence can thus be suitably edited.

Author's Response:

We do agree with the referee that the D_h values should be larger after the UV/vis cycles relative to those of the initial state. Although Figure 4a shows that the D_h value after the first cycle is slightly larger than the initial one, the second cycle affords a D_h value that is almost identical to the initial one. We performed these experiments several times, and the results do not always afford increased D_h values after the UV/vis cycles. Even though a small deviation of less than 50 nm should be associated with this DLS experiment, the DLS results for the folding/unfolding (after UV or vis irradiation) are reproducible.

Specific Comments (4):

(i) DLS has been done with further dilution to avoid cross-linked structures. How good is the kinetic stability of the individual supramolecular chains to ensure that this dilution does not affect the analyses! (ii) Does the Number % in DLS analyses also give a similar trend?

Author's Response:

Please allow us to address the comments separately.

(i) We used AFM (results given below) to examine any potential changes in the degree of polymerization occurs upon dilution and the results does not exhibit any significant change neither in length nor loops of the chains. The results suggest good kinetic stability.

Review-only-Figure 2. AFM images of $\text{SP}_{\text{spiral}}$ at a concentration of 1.0×10^{-4} M (a) and upon dilution to 2.5×10^{-5} M (b). For these two experiments, 10 μL (a) and 10 $\mu\text{L} \times 4$ (b) of $\text{SP}_{\text{spiral}}$ solutions were injected onto the HOPG (5×5 mm), respectively. Scale bars: 100 nm.

(ii) The number % analysis in the DLS experiments did not show the expected photoinduced size changes. This may be due to the fact that the number % analysis of the DLS data may not be applicable to particles with inhomogeneous size distributions and/or non-spherical particles. In this context, please also see:

<http://www.materials-talks.com/blog/2014/01/23/intensity-volume-number-which-size-is-correct/>

Specific Comments (5):

I am very curious to see whether mixed rosettes can be made with five molecules of **1** and one molecule of **2**. Do they self-sort or co-assemble. If they co-assemble it would be interesting to see a similar conformational control! However this study is certainly not required for the present manuscript.

Author's Response:

We thank the referee for his/her constructive suggestion. We have already attempted co-aggregation of **1** and **2** in order to achieve similar conformational control. However, under conventional mixing conditions, i.e., slow or fast cooling of the molecularly mixed state, we only observed self-sorting. This should be due to the large difference in aggregation power of these molecules, as well as their different packing structures.

REVIEWERS' COMMENTS:

Reviewer #1 (Remarks to the Author):

The authors have addressed all remarks adequately, although the number of changes in the text are limited. I can recommend publishing the version as it is now. Especially my comments (ref 1) are made it to the text, but many of the others did not and were only explained in the comments. I could see the addition at a few places with a sentence or a note to address some of the questions of the reviewers 2 and 3, as many readers could have the same questions. But this issue is solved as soon as the referee reports and answers of the authors are coming on line. Despite this, I can recommend this beautiful piece of work.

Reviewer #2 (Remarks to the Author):

I am pleased with the revision by the authors. My and the other referees' concerns have been largely answered.

Only with regard to the use of the correct nomenclature the authors intentionally do not want to change their wording ("We appreciate the referee's reservations, but in the present study, we would like to use the terms "conformation" as well as "folding" and "unfolding" for our supramolecular polymers due to the following reasons: ..."). The reasons given are not convincing and just because other people carelessly use these words, does not make it better. Instead they should have thought of alternative wording. For example, "folding" should correctly be described as "aggregation mode involving non-nearest neighbor interactions" while "unfolding" refers to an "aggregation mode without interactions between supramolecular chains". Perhaps, it is the best compromise if the authors when using the words "conformation" and "folding/unfolding" for the first time make a comment (either in parenthesis or as footnote): "Although these words ... are commonly reserved for covalent polymer chains, we are using them here in the context of non-covalent polymer chains to illustrate the conceptual similarity."

Once such note has been inserted, I feel the manuscript is ready for publication.

Answer to the comments raised by referee 1

Reviewer's Comments:

The authors have addressed all remarks adequately, although the number of changes in the text are limited. I can recommend publishing the version as it is now. Especially my comments (ref 1) are made it to the text, but many of the others did not and were only explained in the comments. I could see the addition at a few places with a sentence or a note to address some of the questions of the reviewers 2 and 3, as many readers could have the same questions. But this issue is solved as soon as the referee reports and answers of the authors are coming on line. Despite this, I can recommend this beautiful piece of work.

Author's Response:

Thank you very much for recommending for acceptance as it is now and we are delighted to see his/her comment "beautiful piece of work". Great to see that referee 1 is satisfied with the changes made against his/her queries, and referee 2 is satisfied with overall changes made while making a comment "My and the other referees' concerns have been largely answered.". We agree that some explanations were in the comments section only as there was almost no scope to include them in the manuscript.

In response to the comment "this issue is solved as soon as the referee reports and answers of the authors are coming on line.", we happy to inform you that we are going to publish the reviewer comments to the authors and author's rebuttal letters online as an additional supplementary peer review file, in line with the newly launched scheme of the journal.

Answer to the comments raised by referee 2

Reviewer's Comments:

I am pleased with the revision by the authors. My and the other referees' concerns have been largely answered.

Only with regard to the use of the correct nomenclature the authors intentionally do not want to change their wording ("We appreciate the referee's reservations, but in the present study, we would like to use the terms "conformation" as well as "folding" and "unfolding" for our supramolecular polymers due to the following reasons: ..."). The reasons given are not convincing and just because other people carelessly use these words, does not make it better. Instead they should have thought of alternative wording. For example, "folding" should correctly be described as "aggregation mode involving non-nearest neighbor interactions" while "unfolding" refers to an "aggregation mode without interactions between supramolecular chains". Perhaps, it is the best

compromise if the authors when using the words "conformation" and "folding/unfolding" for the first time make a comment (either in parenthesis or as footnote): "Although these words ... are commonly reserved for covalent polymer chains, we are using them here in the context of non-covalent polymer chains to illustrate the conceptual similarity."

Once such note has been inserted, I feel the manuscript is ready for publication.

Author's Response:

We would like to thank the reviewer.

According to the suggestion made by the learned referee, we added the following sentence at the end of the introduction section. "Although the terminologies conformation and folding/unfolding are commonly reserved for covalent polymer chains, we are using them here in the context of non-covalent polymer chains to illustrate the conceptual similarity." Please note that we placed the sentence without parenthesis as we thought that the sentence fits with the context.